# Associations between Developing Sexuality and Mental Health in Heterosexual Adolescents: Evidence from Lower- and Middle-Income Countries—A Scoping Review

Neelam Saleem Punjani [1,*](ID), Elizabeth Papathanassoglou [1], Kathleen Hegadoren [1], Saima Hirani [2](ID), Zubia Mumtaz [3](ID) and Margot Jackson [4]

1   Faculty of Nursing, University of Alberta, Edmonton, AB T6W 2K1, Canada; papathan@ualberta.ca (E.P.); kathy.hegadoren@ualberta.ca (K.H.)
2   Faculty of Nursing, University of British Coumbia, Vancouver, BC V6T 2B5, Canada; saima.hirani@ubc.camarg
3   School of Public Health, University of Alberta, Edmonton, AB T6G 1C9, Canada; zubia.mumtaz@ualberta.ca
4   Faculty of Nursing, MacEwan University, Edmonton, AB T5K 2Y9, Canada; jacksonm5@macewan.ca
*   Correspondence: npunjani@ualberta.ca

**Abstract: Background:** During puberty and emerging sexuality, adolescents experience important physical, mental, and social transformations. In the process of dealing with these changes, adolescents can become potentially vulnerable to mental health problems. **Aim:** The aim is to identify and synthesize published research evidence on sexuality-related mental health stressors among adolescent girls and boys, identify gaps (if any) in the current knowledge, and contribute to the knowledge about the experiences of emerging sexuality and health among adolescents, to further inform research, practice, and policy initiatives in sexual health. **Design:** A scoping literature review of peer-reviewed articles published between 1990 and 2021. MEDLINE, CINAHL, EMBASE, PsycINFO, Global health, ERIC, and Sociological Abstracts databases were searched for research studies that reported experiences of sexuality-related mental health issues and symptomatology of adolescents. We targeted studies conducted with adolescent populations between ages 10–19 years living in LMICs. **Results:** Data from 12 published research papers, including 8 qualitative studies, 3 quantitative studies, and 1 mixed method study, were systematically analyzed. Four major themes and four sub-themes were identified regarding the sexual and mental health of adolescents: (1) Relationship of sexuality and mental health; (2) Social and cultural influences; (3) Challenges in seeking sexuality information and services among adolescents; and (4) Educational needs among adolescents related to sexuality. **Conclusions:** Lack of social support, unmet needs for accessible adolescent-friendly sexual health services, counseling, and age-appropriate information may be associated with several mental health stressors and symptoms, such as sadness, depressive and anxiety symptomatology, regret, fear, embarrassment, low self-esteem, guilt, shame, and anger. Therefore, tackling sexuality-related stressors could play an important role in addressing the overall well-being of young people. Future studies need to generate a deeper understanding of the concept of sexual health and its relation to mental health in diverse contexts. **Implications for Practice:** Health care professionals need to be aware of sexuality-related experiences of adolescent girls and boys by offering effective youth-friendly sexual and reproductive health education to support overall mental health and improve the experiences of emerging sexuality in adolescents.

**Keywords:** sexuality; mental health; adolescents; stressors; low- and middle-income countries

## 1. Introduction

### 1.1. Background

Adolescence is a critical period in the transition from childhood to adulthood, during the course of which adolescents aged 11 to 19 years take on new responsibilities and

experiment with independence [1–3]. A great deal of research on this transitional period exists, in terms of physical, cognitive, psychosocial, and interpersonal development and how these developmental aspects affect adolescents' mental health and well-being [2]. One of the less well-studied processes is adolescents' emerging sexuality and the development of the sexual self in the context of family, community, and society. The generalizability of this knowledge to other contexts from researchers who have explored this process depends on the country, culture, and social norms [2–5]. Thus, it is important to add to this body of knowledge by exploring the development of sexuality across multiple adolescent populations across countries and cultures. As an initial step, this scoping review will highlight our understanding to date and identify current gaps in the literature.

The definition of sexuality has evolved over time [6–9]. However, for the purposes of this scoping review, the WHO's comprehensive and gendered description of sexuality guides us: "Sexuality is a central aspect of being human throughout life encompasses sex, gender identities and roles, sexual orientation, eroticism, pleasure, intimacy and reproduction. Sexuality is experienced and expressed in thoughts, fantasies, desires, beliefs, attitudes, values, behaviours, practices, roles and relationships. While sexuality can include all of these dimensions, not all of them are always experienced or expressed. Sexuality is influenced by the interaction of biological, psychological, social, economic, political, cultural, legal, historical, religious and spiritual factors" [6] (para. 6). We focused on gender as in many low- and middle-income countries (LMICs) traditional gender roles shape the way adolescent girls and boys explore their sexualities.

A greater percentage of the population in developing countries is young compared to that of the developed countries of the world [4]. According to the United Nations Population Fund ((UNFPA), today's cohort of young people aged 10 to 24 years is the largest in history; they number over 1.8 billion, and 90% live in low- and middle-income countries (LMICs). A large number of young girls and boys around the world are sexually active, and this percentage rises steadily from mid-to-late adolescence [10,11]. Globally, 11% of childbirths and 14% of maternal mortality involve 15- to 19-year-old girls, and 95% of adolescent births occur in developing countries [11–13]. Annually, 4 million adolescent girls have unwanted pregnancies [11], and 3 million adolescent girls undergo unsafe abortions [14]. Worldwide, among people who live with human immunodeficiency virus (HIV), 1,300,000 are adolescent girls and 780,000 are adolescent boys [15,16]. Even though many countries have emphasized their commitment to eradicating early marriage, the tradition remains in numerous countries of the world. Early marriage corresponds with the prohibition of girls' rights to choose whom and when to marry [17–19].

Thus, the data on adolescents' sexual activity can be difficult to interpret because of the early age of marriage in some LMICs. The emerging sexuality that accompanies puberty can cause challenges for adolescents [20,21], which arise from adapting to changes in appearance and the functioning of a sexually maturing body, dealing with sexual desires, encountering varied sexual attitudes and values, and desiring to experiment with certain sexual behaviors. Moreover, incorporating these feelings, attitudes, and experiences into a developing sense of self adds further challenges for adolescents [20,22,23]. Adolescents who live in LMICs may be at an increased vulnerability to their social and environmental situations, for instance, strict socio-cultural norms, violence, and barriers to access to health care services [24,25]. The social and cultural context in which young people live greatly influences their responses to these challenges.

Psychosocial stressors have been linked to mental health issues such as depression and anxiety symptoms [22,26,27]. Adolescent girls and boys are also potentially at risk for participation in risky sexual activities, substance abuse, and violence associated with their psychological well-being and mental health [22,26]. The consequences of risky sexual behavior can be unintended pregnancy and sexually transmitted infections (STIs), including HIV infection [22].

Although we assume a bidirectional link between adolescent sexuality and mental health, very limited literature exists on how sexuality-related issues influence the psycho-

logical well-being of the adolescent population in LMICs. The majority of the published literature on adolescents' sexuality has focused on sexual activity and its consequences; very little has addressed the mental health aspects of sexuality [28]. Furthermore, most existing studies from LMICs on adolescent sexuality have explored their physical rather than their psychosocial experiences during adolescence [22]. There is a paucity of information regarding associations between developing sexuality and mental health in adolescents in LMIC. Given this knowledge gap and the multidimensional nature of sexual and mental health, a scoping review is ideal to determine the volume and nature of the literature, as well as the current state of knowledge.

*1.2. Aim*

The aim of this scoping review is to describe, evaluate, and synthesize published evidence on sexuality-related mental health stressors among adolescent girls and boys in LMIC, to identify gaps (if any), and to contribute to the knowledge about the experiences of emerging sexuality among adolescents. The ultimate goal is to inform research, practice, and policy initiatives on the associations between sexual and mental health.

## 2. Materials and Methods

### 2.1. Methodology

A methodical approach was used to guide the analysis of both theoretical and empirical literature to generate a comprehensive understanding of experiences of emerging sexuality and related mental health stressors among adolescents. Using this form of knowledge synthesis allows for the broad exploration of adolescent sexuality and mental health to map key concepts, evidence types, and gaps in research in a defined field. Furthermore, a scoping review makes use of a wide array of knowledge exhibited through empirical research and anecdotal accounts [29,30]. This type of review process can add to the rich contextual component, which is necessary for the exploration of adolescent sexual and mental health in a broader context.

This scoping review employed the methodological framework proposed by Arksey and O'Malley (2005) and further refined by Levac, Colquhoun, and O'Brien (2010), and the Joanna Briggs Institute [31]. A scoping review methodology was chosen since the area of adolescent sexuality and its association with psychological well-being has not been reviewed comprehensively before in the context of LMICs. This methodology is particularly useful for examining a broadly covered topic, in order to comprehensively, systematically map the literature, and identify key concepts, theories, evidence, or research gaps [29,30,32]. The research question guiding this scoping review was: "What are the experiences of developing sexuality and its potential associations with mental health among adolescent girls and boys living in LMICs?" The research question was developed using the SPIDER (Sample, Phenomenon of Interest, Design, Evaluation, Research type) format as recommended by Cooke, Smith, and Booth (2012), as a relevant method for structuring qualitative research questions [33] (Table 1). The use of the SPIDER method helped to refine the question and ensured that the appropriate evaluation measures were employed.

**Table 1.** Components of the Research Question using the SPIDER Format.

| Spider Tool | Justification |
|---|---|
| S—Sample | Adolescents (10–19 years) |
| PI—Phenomenon of Interest | Experiences of developing sexuality and associated mental health issues/psychological well-being |
| D—Design | Qualitative, quantitative, mixed methods |
| E—Evaluation | Experience and perceptions of adolescent girls and boys |
| R—Research type | Mixed method designs |

The scoping review method includes six stages: (a) formulating the research question; (b) identifying relevant studies; (c) selecting the literature (an iterative process); (d) charting the data; (e) collating, summarizing, and reporting the results; and (f) developing a knowledge translation plan and consulting interested stakeholders.

### 2.2. Data Sources and Search Strategy

MEDLINE, CINAHL, EMBASE, PsycINFO, Global health, ERIC, and Sociological abstracts databases were searched for research studies that focused on the experiences of the sexuality-related mental health stressors of adolescents. With the assistance of a librarian, the following search key terms were mapped using the SPIDER tool and were used to locate pertinent articles: sexuality/sex/sexual health, mental health issues/stressors/anxiety/depression/psychological well-being, adolescents/teenagers/youth, puberty, and low- and middle-income countries/developing countries (Table 2). The search terms were recorded for review purposes and for maintaining transparency of the selection process.

**Table 2.** Mapping Keywords using the SPIDER Tool.

| Spider Tool | Search Terms |
|---|---|
| S—Sample | "young" OR "teen*" OR "youth*" OR "low and middle-income countr*" OR "Developing countr*" OR "South Asia" OR "low and middle-income countr*" |
| PI—Phenomenon of Interest | "sex" OR "sexual health" OR "sexuality" OR "mental health" OR "puberty" OR "stress*" OR "anxiety" OR "mental disorder* OR "depress*" OR "psychological well-being" |
| D—Design | "questionnaire*" OR "survey*" OR "interview*" OR "focus group*" OR "case stud*" OR "observ*" |

### 2.3. Study Eligibility Criteria

As recommended by Levac, Colquhoun, and O'Brien (2010), inclusion and exclusion criteria were developed at the beginning of the scoping process. These criteria served as a guide for the reviewers on which we based a decision about the literature to be included in the scoping review.

Inclusion criteria comprised: (a) primary qualitative, quantitative or mixed methods studies addressing associations between sexuality and psychological well-being or mental health; (b) studies conducted in LMICs with adolescents and young adults (11–24 years of age); (c) articles written in English; and (d) articles published between 1990 and 2021. Although the adolescent population is defined as aged 10–19 years, since many studies target youths aged 15–24 years along with adolescents, we extended the inclusion criteria to include studies that involved young adults along with adolescents.

Exclusion criteria included studies referring to lesbian, gay, bisexual, queer, and transgender population specifically, and those focusing exclusively on sexual abuse, HIV, rape, violence, homelessness, and substance users. These population are particularly excluded because of the susceptibility of social exclusion and peer victimization that they face and such victimization has well-documented psychological consequences that could have compromised the results of the scoping review. Various filters were used to remove duplicates and those that did not meet the inclusion criteria (written in a language other than English and publication dates outside set parameters). In addition to the databases, a hand search of various articles was carried out in order to identify references specific to experiences of adolescent sexuality and related mental health stressors among adolescent girls and boys.

### 2.4. Data Extraction and Synthesis

All the articles were read in detail and data were extracted from the methods and results section. Data on the types of sexuality-related stressors, associations between

sexuality and mental health, and experiences of adolescent girls and boys during puberty were extracted and analyzed. Findings from each article were summarized in a table format and content analysis was performed to extract major themes. A descriptive synthesis table was formulated containing the textual descriptions of all the findings (Table A1). Extracted data were grouped together and clustered into categories to formulate themes and subthemes. Conceptual mapping was performed to identify the relationships within and between study characteristics and results.

Content experts in the areas of sexuality, adolescent, and mental health were involved to generate critical reflections throughout the review process and to obtain a consensus over the generated themes. Figure 1 summarizes the literature review process—PRISMA diagram. All data were extracted by the lead author in consultation with the two co-authors.

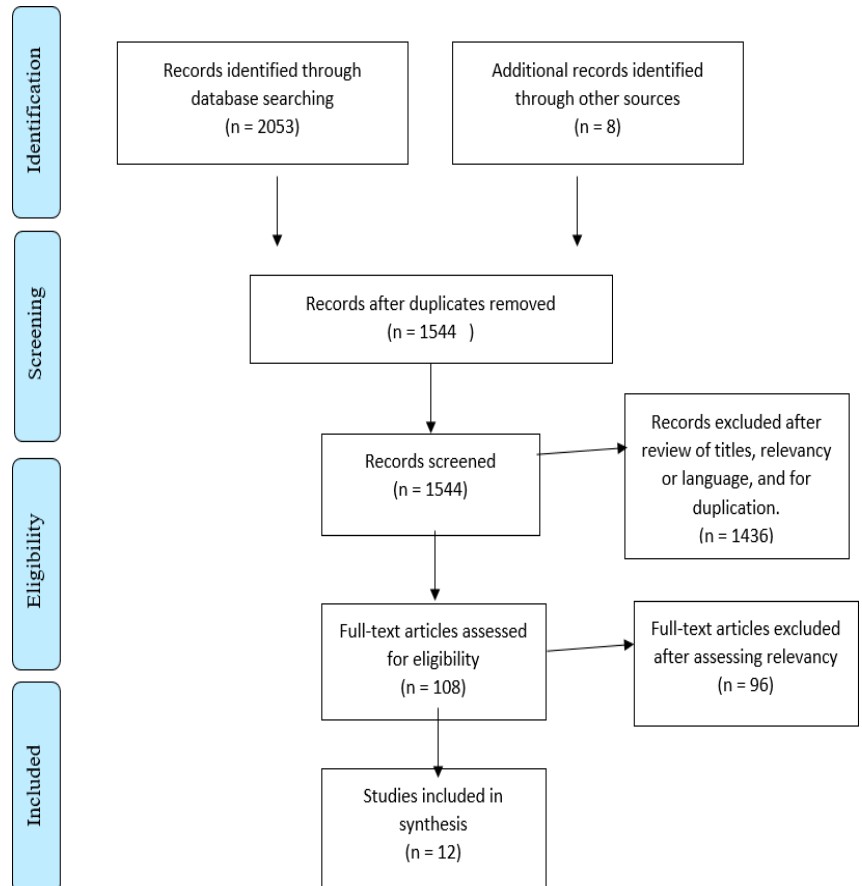

**Figure 1.** Literature Search Process: PRISMA Diagram.

### 3. Results

*3.1. Characteristics of Identified Studies*

The initial search retrieved a total of 2061 articles. After removing duplicates and articles in other languages and comparing abstracts against eligibility criteria, a total of 1544 studies were considered relevant. After reviewing title and abstracts, 1448 studies were excluded, as they did not meet eligibility criteria (not relevant to the main subject, study population were adults, full texts were not available, and language other than English). Following a full-text review of 96 articles and consultation among the authors, 12 articles were included in the final review and analysis. Twelve studies examined the associations between sexuality-related stressors and mental health. A summary of the identified articles is provided in Table 3.

**Table 3.** Assessment of Methodological Quality According to the Mixed Methods Appraisal Tool (MMAT, Pluye et al., 2011).

| Study Design | Selected Studies | Appraisal Score |
|---|---|---|
| Qualitative Studies | Crichton, Kenya, 2013 | 100% (****) |
| | Aziato et al., Ghana, 2016 | 50% (**) |
| | Girod et al., Kenya, 2017 | 100% (****) |
| | Joan et al., Malaysia, 1998 | 50% (**) |
| | Agampodi et al., Sri Lanka, 2008 | 50% (**) |
| | van Reeuwijk, Bangladesh, 2016 | 100% (****) |
| | Bello et al., Kenya, 2017 | 100% (****) |
| | Lahme et al., Zambia, 2018 | 75% (***) |
| Quantitative Studies | Kyagaba et al., Uganda, 2014 | 100% (****) |
| | Khopkar et al., India, 2017 | 75% (***) |
| | Ramathuba, South Africa, 2015 | 75% (***) |
| Mixed Method | Biney, Ghana, 2016 | 100% (****) |

** (50% of quality criteria met); *** (75% of quality criteria met) or **** (100% of quality criteria met).

The majority of studies used a qualitative descriptive design (*n* = 8) [34–40], a few studies used a quantitative design (*n* = 3) [41–43], and only one study used a mixed method approach (*n* = 1) to examine the association between sexual health and mental health [44]. Sample sizes ranged from 11 to 1954 adolescent girls and boys depending on the research design. Some studies included adolescent girls and boys, parents/ caregivers, and health care providers in the sample, but this review focused on findings related to sexuality and sexuality-related mental health issues among adolescents' girls and boys only.

The majority of studies were from the African regions (*n* = 8) and the rest were from Asia (*n* = 5). The studies from Africa were based out of Ghana, Kenya, Uganda, Nigeria, Zambia, and Limpopo. The studies from Asia were conducted in India, Malaysia, Sri Lanka, and Bangladesh. All the studies were conducted with populations between the ages of 11 and 24 years. Overall, most studies included predominantly girls as participants (*n* = 8), and the rest (*n* = 4) both male and female participants. However, it was not possible to perform a gender analysis as gender-specific results in terms of girls and boys were not reported in most of the primary studies. Out of 12 studies, 7 employed purposive sampling, 3 studies convenience sampling, and only 2 studies used random sampling. The qualitative studies employed interviews and focus groups as a method of data collection. The quantitative studies employed self-administered questionnaires and surveys as methods of data collection. The qualitative studies were descriptive in nature and therefore lacked an in-depth interpretation of sexuality, experiences of sexuality, and a discussion of its relation to constructs related to mental health.

Of the 12 articles, 8 did not specify mental health as a central focus of their study, but included it as one of many experiences among adolescents in experiencing sexuality. Mental health symptoms were mentioned in all the articles, and the authors addressed different manifestations of mental health issues based on what they felt best served the purpose of their study. This introduced a challenge with respect to deducing the meaning of mental health stressors related to sexuality in these studies.

*3.2. Quality Assessment*

The methodological quality of each included study was appraised by the Mixed Methods Appraisal Tool (MMAT) developed by Pluye et al. (2011) [45]. The MMAT quality assessment process involved answering four questions that are appropriate to the study design regarding recruitment, randomization (if applicable), appropriateness of outcome measures, and attrition rate/completeness of data. Studies were scored using MMAT as

follows: _(0% of quality criteria met); * (25% of quality criteria met); ** (50% of quality criteria met); *** (75% of quality criteria met) or **** (100% of quality criteria met). Nine studies met at least 75% of quality criteria, whereas three studies met 50% of quality criteria. This tool was used to evaluate the quality of studies included, but no studies were excluded from the review based on low quality scores. Most quantitative studies had limitations with regard to external validity, and the validity and reliability of the study instruments used. Most qualitative studies reviewed were descriptive in nature and therefore did not provide an in-depth interpretation of the findings.

*3.3. Major Themes*

The scoping review generated four major themes and a few sub-themes: (1) Relationship of sexuality and mental health; (2) Social and cultural influences; (3) Challenges in seeking sexuality information and services among adolescents; and (4) Educational needs among adolescents related to sexuality (Figure 2).

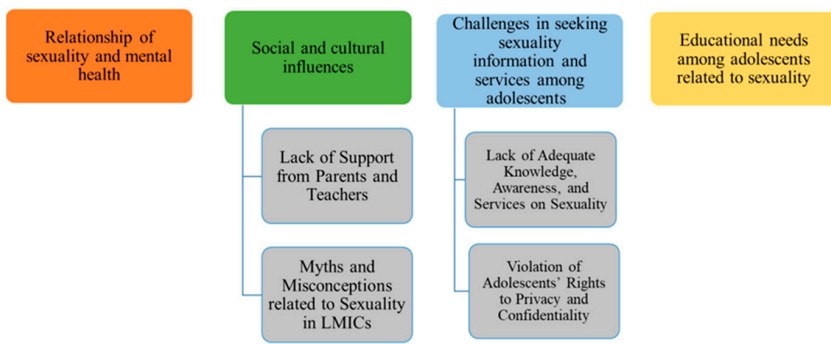

**Figure 2.** Diagram showing Themes and Sub-Themes.

*3.4. Relationship of Sexuality and Mental Health*

In the identified studies, adolescent sexuality was often implicated in mental health responses as reported by the participants. Although most adolescents may experience alterations in mood, regardless of sexuality, the identified studies support that adolescents and youth (11–24 years) experience altered mood and emotions in relation to developing sexuality [34,41,42]. These included sadness, depression, regret, fear, anxiety, embarrassment, low self-esteem, guilt, shame, and anger [34,36–44,46]. If such responses persist, they may constitute risk factors for impaired mental health. The adolescent period was expressed as a time when young girls and boys develop strong curiosity about the male and female body, virginity, sexual intercourse, menstruation, masturbation, sexual power, and sexually transmitted diseases [40].

Most of the stressors that were associated with sexuality were related to the lack of knowledge and incomplete information about the physical change, the lack of preparedness to have sex, fear and worry about family members being aware of their sex lives, and unmet medical and sexual health counselling needs [36,37,40,44].

The studies suggested that psychological distress is experienced due to various reasons regarding the menstrual cycle and masturbation. For example, Biney (2016) examined the relationship between adolescents' self-concept and their sexual and mental health among Ghanaian youths and found that even the highly sexually confident adolescent girls and boys also exhibited poorer mental health [44]. Particularly for girls, unlike the other gradual changes that accompany puberty, menarche was considered as a sudden and conspicuous change and provides a dramatic demarcation between girlhood and womanhood [38,41,46]. In addition, studies found that school-going adolescent girls face fear of security and harassment from boys while menstruating, which is harmful to their mental health [38,39]. Young girls reported discussing sexual health matters with friends in order to seek help. On the other hand, boys do not discuss and report their sexual health issues with anyone due to lack of trust [38,39].

### 3.5. Social and Cultural Influences

Sociocultural norms may prescribe that unmarried adolescent are sexually inexperienced and ignorant [47]. As a result, when adolescents face any sexual health problems, it is difficult for them to reveal the situation to family members, friends, and health care providers, who rarely respect their confidentiality [48]. Aziato et al. (2016) used a vignette-based focus group approach to have adolescents reflect on a scenario related to unwanted pregnancy [35]. In reflection, the respondents believed that a pregnant adolescent would not tell her mother because the mother would be angry, mad, disappointed, unhappy, hurt, worried, disgraced, ashamed, and sad. They also thought that the mother would shout at, beat, or sack her (tell her to leave the house). The studies also found that due to socio-cultural contexts of LMICs, many girls were unable to obtain accurate information about what menstruation is and how to hygienically manage menstruation [38,40]. van Reeuwijk and Nahar (2013) found that, due to socio-cultural influences in Bangladesh, young girls and boys feared involvement in any kind of romantic relationship. Despite the strong interest and desire of young people to have love affairs, they were taught not to indulge in such activities as it will damage the reputation of their families. Hence, it will hamper their future marriage prospects [40]. Girls also reported wearing a burkha (veil) while dating, as they are afraid of being caught by family members [40].

Crichton et al. (2013), Lahme et al. (2018), and Ramathuba (2015) also found that cultural and social taboos and initiation ceremonies related to menstruation had detrimental implications for girls' emotional well-being [37,39,43]. A study by Crichton et al. (2013) in Nairobi provided testimony that the negative emotional and psychosocial impacts of menstruation due to social stigma were an important concern for girls and involved fear of stigma and feelings of embarrassment, anxiety, and low mood [37]. Overall, the mental health consequences of sexuality-related stressors in resource-constrained countries in Asia and Africa remain unacknowledged.

### 3.6. Lack of Support from Parents and Teachers

Open discussion of sexual matters in households, schools, and public places for adolescents is inhibited by personal embarrassment, and also by conservative social norms and religious prescriptions [49]. This situation leaves many adolescents with insufficient knowledge and skills to manage sexual health. Crichton et al. (2013) and Khopkar et al. (2017) found that adolescent psychological well-being was associated with a lack of information and guidance regarding sexuality from parents [37,41]. Crichton (2013) reported that many girls in their study in Nairobi described "psychological deprivation" due to lack of access to accurate information regarding sexuality, and limited emotional and practical guidance and support [37] (p. 902). Many of the girls had heard about menstruation from family members, teachers, or friends before menarche, but the information they received was often vague or inaccurate [37]. Chrisler and Zittel (1998) also found that 22% of Sudanese girls in their study were completely unprepared for menstruation and did not know what it was when they first experienced it [36]. The participants in the study reported that "their mothers or aunts had lied to them about menstruation, deliberately giving incorrect information, which increased their sense of being unprepared for the reality of menarche" (p. 306). Moreover, one girl mentioned in story that her aunt told her, "Don't flash your smiles to boys or else you might get pregnant" [36] (p. 310).

Likewise, very little attention may be paid to adolescent girls' sexuality needs in school. Girod et al. (2017) found that adolescent girls in Nairobi, Kenya, once they started menstruating, were told by their schoolteachers to be aware of men and boys, implying that the girls were now sexual objects and that boys were not accountable for their actions [38] (p. 843).

This discourse may lead girls to believe that they are at fault if they are harassed, which may have serious mental health consequences [38]. Similarly, in a study in Zambia, girls reported that they were intimidated and embarrassed by the treatment they received from their male teachers and peers who teased, mocked and humiliated them, therefore

they often opted to stay at home when menstruating, rather than exposing themselves to this harassment [39].

### 3.7. Myths and Misconceptions Related to Sexuality in LMICs

The available data suggest the existence of several sexual health myths and misconceptions in adolescents. Most existing studies indicate that adolescents expressed a multitude of feelings in relation to sexuality, most notably, curiosity, desires and pleasure, and feelings of insecurity and concerns [40,46]. Adolescent girls expressed misconceptions and curiosity in relation to virginity, menstruation, sexual power, sexually transmitted diseases, and homosexuality [40,42,44]. Many girls expressed worries and various misconceptions about the issue of virginity and were insecure about their ability to prove their own virginity. Some older girls wanted to know of ways a girl could increase her sexual power to please men. The word "sexual power" was used to express concerns or ideas about the ability to perform sexually and give/obtain pleasure [40,42,44]. The girls were curious about this because they felt insecure about their ability to please their future husbands sexually. They related this to the concern that if a wife cannot satisfy her husband, the husband will seek a sex worker or have extramarital sexual relationships [40,42,44]. In addition, many girls expressed misconceptions that a girl could become pregnant after kissing or hugging a boy [40]. Chrisler and Zittel also reported that parents or family members perpetuated similar misconceptions [36].

On the other hand, boys had a strong curiosity about the female body and sexual intercourse [40]. Adolescent boys expressed many myths and misconceptions about masturbation and wet dreams and about the size and shape of the penis and duration of intercourse [40]. Curiosity about sexuality among adolescent girls and boys seemed to be driven by the insecurities, concerns, myths, and misconceptions that young people have about their own bodies and (future) ability to perform sexually [36]. Their insecurities and concerns, in turn, encouraged adolescents to look secretively for sources of information on sexuality, for which they relied mainly on peers, the media (erotic books, music, and films) and (for boys) street canvassers [34]. In Bangladesh, street canvassers are con-artists who sell medicines (from biomedical and herbs to amulets) in the streets [40]. They are known for their charismatic way of selling and are very popular, predominantly with men and boys, as it is culturally inappropriate for women and girls to stand among men, while someone is talking about sexuality issues. The canvassers provide a range of unrealistic and false information regarding the size of the penis, duration of intercourse, masturbation, wet dreams, menstrual pain, signs and symbols of virginity, shapes of breasts, the hymen and so on [40]. Street canvassers make use of misconceptions and exaggeration to sell their medicines, which they say will increase sexual power and penis size.

### 3.8. Challenges in Seeking Sexuality Information and Services among Adolescents Lack of Adequate Knowledge, Awareness, and Services on Sexuality

The available studies indicate that, in general, knowledge about sexuality, sexual health-promoting behaviors, and safer sex practices were low among adolescents who reside in LMICs [34–36,39,41,43,44,46]. For example, many adolescents described a lack of reliable access to sexuality information as a major cause of physical and psychological discomfort, embarrassment, anxiety, fear of being stigmatized, and low mood [34–36,39,41,43,44,46]. To describe the emotional distress participants experienced related to sexuality, they used language, such as "feeling bad", "feeling stressed", or "fearful", and "wanting to cry" [37].

Many young adolescents have pointed out how that the lack of knowledge on sexuality led to a lack of confidence in solving sexuality-related issues [34,36,43]. Young people mentioned that their parents and their teachers deliberately made information on sexual health unavailable to them [34,36,43]. Young boys and girls also mentioned that most of the sexual-health-related issues would have been avoided if they were all well-given access to sexual health knowledge [34]. They felt that these topics were usually kept away from them in school and libraries. Even though they had some sources of information, they were not

exposed to them at the appropriate age [34]. In most of the studies, knowledge of available services on sexual health was very limited among adolescents. The lack of knowledge about sexual-health-related matters led to poor self-confidence among adolescents to discuss sexual health matters [34]. Lahme et al. in their study in Zambia, found that neither parents nor teachers provided information to pre-menarche girls or psychological support when they started menstruating, at the time they needed it most [39]. Although studies reported that adolescents are happy to accept sexual health services through public clinics and other health infrastructure, they also demand separate youth-friendly services and health care providers that can ensure their privacy [34,35,46]. Likewise, Kyagaba et al. assessed the unmet sexual health counseling needs among Ugandan University adolescent students (*n* = 2706) and found that unmet sexual health care needs were associated with poor mental health, experience of sexual coercion, and poor self-rated health [42].

### 3.9. Violation of Adolescents' Rights to Privacy and Confidentiality

Bello et al. (2016) reported that a desire for privacy as a response to pubertal body changes increase in adolescents, and the absence of privacy could cause stress and fear. Some studies draw attention to the lack of confidence and trust related to sexuality among adolescents, which may be accentuated by poor adherence to privacy and confidentiality principles by health care providers, teachers, and parents [34,35]. According to Agampodi et al. (2008), one participant from Sri Lanka highlighted concerns about a doctor or nurse asking embarrassing questions in front of their mother, she mentioned that the "Doctor asked me embarrassing questions in front of my mother [34]. As soon as we left the place she started asking me various questions with a tone of blaming; I decided not to seek medical advice again and not to tell anything to my mother" (p. 5). This lack of confidentiality and privacy hinders adolescents from seeking professional help. In the study by Agampodi et al., a 17-year-old boy reported lack of trust in health care professionals in accessing sexual and reproductive health services by saying that "No one cares about boys, but we have problems to discuss. We don't know whether these health care workers are good at solving our problems. The way they treat other illnesses made me feel uncomfortable to discuss sensitive reproductive issues with them" [34] (p. 5).

Confidentiality is the foundation of the therapeutic relationship with young people. However, health care professionals were not always seen as a source of support or unbiased advice; in fact, adolescent girls suggested that they are likely to be judgmental and disrespectful [35]. Many adolescents in a study conducted in Ghana expressed distrust and doubt about health care providers and thought that most nurses, upon hearing that an adolescent was pregnant, would insult her and would tell other nurses about the pregnancy. A few participants also reported that some nurses would hit the adolescent during the delivery of the baby if she carried the unwanted pregnancy to term [35].

### 3.10. Educational Needs among Adolescents Related to Sexuality and Mental Health

The identified studies suggested a lack of specific trials of interventions to manage mental health aspects of sexual and reproductive health problems in adolescents in resource-constrained countries. However, the authors have made general recommendations to improve adolescents' psychological well-being related to sexuality. According to Khoopkar et al., to ensure health promotion among adolescents, health care organizations should provide integrated mental health along with other sexual and reproductive health services. Moreover, the authors suggested that youth centers and community centers should have the means to provide professional education and counseling on the culturally sensitive topic of sexual health for adolescent girls and boys [41]. Aziato et al. emphasized that more effort is needed to train health care providers to help adolescents to figure out their options in a safe and unbiased manner [35]. In addition, school nurses should be made available to assist in enforcing sexuality education amongst female students with regard to menstruation, sex, teenage pregnancy, conception and contraception [39,43]. Bello et al. also emphasized the importance of building the capacity of parents for effective sexual

health-related communication with their young children before and during their pubertal years [46].

Studies concluded that more attention is needed to girls' early experiences of menstruation, as misinformation related to menstruation increases negative stereotypes, which leads to poor mental health [36]. Ramathuba suggested that maturing girls should be empowered to view menstruation as a normal physiological process and not to shy away or keep it a secret for fear of embarrassment amongst peers if they matured early [43].

Biney proposed that there is a need to acknowledge the role of sex education for both adolescents and adults in the community, in order to promote the sexual health of young people [44]. One of the recommendations by van Reeuwijk and Nahar was to develop interventions by using modern media, as it is a popular source among adolescents to access sources of information on sexuality [40]. Additionally, the need for curriculum-based programs is echoed as a recommended strategy to improve sexual health of adolescents. It is also important to train schoolteachers to tackle these sensitive issues in order to gain adolescents' confidence by improving their mental health [34].

## 4. Discussion

In this scoping review, we identified 12 primary studies that addressed sexuality-related mental health stressors across LMICs in Asia and Africa. The identified studies offer evidence supporting a role for sexuality-related issues in shaping the mental health of adolescents. The results of this review could help health care professionals who practice in the area of sexual health to understand better the sexuality-related mental health issues and psychological well-being of adolescents, especially in the context of LMICs.

Our findings are in accordance with studies conducted in developed countries, such as the U.S.A. and U.K., showing that pubescent youth are susceptible to poor mental health outcomes because of the dearth of accessible adolescent-friendly health services and restrictions to access to appropriate and accurate knowledge, particularly for unmarried females [50]. Researchers who have conducted studies in LMICs have suggested that the stigma attached to adolescent sexual behavior, unintended pregnancy, early childbearing, abortion, and STIs can result in risky and unsafe behaviors, and unfavorable health and social outcomes. This includes shame, social marginalization, violence, and mental health illness, which further restrict access to sexual health services [51,52]. These findings are parallel to the findings of our review.

The results of this scoping review draw attention to several aspects of sexual health, including privacy, confidentiality, health care services, and sociocultural norms. Sexuality is a sensitive issue in any culture, and the norms that regulate sexual behavior vary from one geographical area to another, from one subculture to another, and even from one age group to another [47]. The lack of open discussion of sexual matters with parents, teachers, and friends because of embarrassment, fear, shame, stigma, and conservative socio-cultural and religious norms contribute to adolescents' inadequate knowledge and skills to manage sexual health issues [53]. For example, we found that menstruation is usually associated with religious and cultural beliefs in Asian and African cultures [36,37], which may create challenges in accessing appropriate health care services and speaking openly about menstruation. The perpetuation of the cultural perspective that menstruation is 'dirty' and that it must be hidden and should not be discussed in mixed company deprives adolescent girls of the opportunity for more information to take control of their sexual health and ensure their psychological well-being. However, studies conducted in four African countries in Burkina Faso, Ghana, Malawi and Uganda have shown that age-appropriate and informed discussions on sexuality between parents and adolescents make the youth in the community more sexually healthy [54,55].

The issue of confidentiality with regard to adolescent sexuality involves careful consideration of how to address adolescents, for example, to ensure the protection of their integrity and respect existing societal values and subculture values [56]. In our review, we found that adolescent girls and boys do not always consider health care professionals as sources

of support or unbiased advice and, in fact, consider them judgmental and disrespectful. Moreover, studies examining attitudes of healthcare providers towards contraceptives for unmarried adolescents and factors affecting the adequate provision of these services to adolescents in Nigeria and Cape Town, South Africa corroborate these findings [57,58].

In accordance with our findings, previous evidence from LMICs also demonstrates that sexual and reproductive health services that target adolescents are extremely disjointed, poorly synchronized, and low in quality [5,59,60]. Additionally, our findings are similar to those of previous reports showing that health care professionals face numerous challenges in providing care to adolescents, because they need specialized skills and knowledge for consultation, interpersonal communication, and interdisciplinary care [11]. This finding is understandable in view of previous studies that emphasized that the attitudes of health care professionals need to change to enable adolescents to seek help from qualified health care providers for safe sexual health practices [60,61]. We also found that training and educating professionals, developing stakeholder interrelationships, and using evaluative and iterative strategies are frequently recommended strategies to introduce and promote change in adolescents' sexual health practices, which is similar to the findings of other studies conducted in Asian and African context [11].

In the sociocultural context of LMICs, sexuality is considered the privilege of older and married individuals, which makes it extremely difficult for young people to access sexual health counseling [62]. Other studies have supported these findings and shown that the stigma of risky sexual behaviors and STIs, including HIV and AIDS, further restrict the access of those who are stigmatized to sexual health services. Families, communities, and the healthcare system can be agents of stigmatization through such behaviors as abusing, insulting, and deserting adolescents [63,64]. Consequently, young people might use withdrawal as a coping strategy in the face of perceived or experienced stigma. This could also explain the finding of a strong association between adolescent girls' and boys' feelings of loneliness and their failure to seek sexual health care when they need it.

The concern about the confidentiality of adolescents' personal information is a substantial hurdle to access to sexual healthcare services. A study conducted in Tanzania with young people has shown that adolescents may have a profound fear that their parents will learn about their accessing sexual health services [65]. In agreement with our findings, previous researchers have shown that adolescents are concerned that family, friends, or other community people who are acquainted with their parents will recognize them in the waiting room. They might also worry that healthcare providers who have social connections with their parent(s) or guardian(s) might purposefully or unintentionally reveal confidential information [66]. Alford found that, if her adolescent participants' healthcare professionals notified their parents, 83% would discontinue access to sexual health services, whereas only 1% would abstain from sex [67].

Adolescent girls and boys often require guidance in making decisions, especially in dealing with sexuality issues. Biddlecom et al. and Namisi et al. offered insight on the importance of sexuality education and recommended that adolescents should receive essential information and learn skills through comprehensive sexual and reproductive health education to prevent mental health problems [54,55]. Biddlecom et al. and Namisi et al. also suggested that age-appropriate and informed discussions on sexuality between parents and adolescents improve the sexual health of youth in the community [54,55]. However, more work is needed in LMICs to ensure that adolescents receive accurate education on sexuality to understand how to practice healthy sexual behaviors eventually.

Our review revealed that the persistent inequality among adolescent girls and boys and restrictive gender norms can be translated into a range of negative mental health outcomes, especially for young girls. These findings are understandable in view of the work of Blum, Mmari, and Moreau in their study in 15 different countries of children aged 10–14 years, they found that boys are constantly encouraged to be strong and autonomous, whereas girls are considered vulnerable and in need of protection. Moreover, with the

onset of puberty, boys are expected to prove their toughness and sexual ability, and girls are responsible for attracting male attention [68].

In addition, their peers persecute and mock boys who do not achieve local masculinity standards, but girls who transgress the social norms of sexual propriety are shamed and humiliated [68,69]. Concerns about female sexuality and reputational risk cause parents to tightly control their daughters' behavior and freedom of movement, which can affect their psychological well-being.

Sexuality embraces so much more than just the physical act and has both physical and psychosocial components (East & Hutchinson, 2013; Hensel, Nance, & Fortenberry, 2016). The ways in which individuals express their sexuality depend on a range of factors, such as culture, religion, society, economics, politics, law, history, and spirituality [8,22].

The current agenda for Sustainable Development 2030 recognizes the need for greater accountability, especially for the Global Strategy for Women's, Children's and Adolescents' Health [70]. Our findings indicate a paucity of research regarding the association between sexuality-related stressors and mental health among adolescent populations. Most of the research that is available has focused on girls, and there is a major gap in knowledge on the experiences of boys. This implies an urgent need for comprehensive research on the relationship between emerging sexuality and mental health in adolescents.

## 5. Limitations

This scoping literature review has several limitations. The findings of this scoping review are not generalizable to settings other than Asian and African LMICs or populations other than adolescents. Moreover, the review included only articles written in English. It is likely that valuable research on sexual health and mental health has been published in other languages. Additionally, the review did not cover sexual health knowledge among diverse groups, such as lesbian, gay, bisexual, transgender, transsexual, and queer (LGBTQ), which could differ from that of the populations described in the primary studies that were included in the review.

## 6. Conclusions

This scoping review identified several sexuality-related mental health issues among adolescent girls and boys in LMICs and their influence on shaping adolescents' overall mental well-being. Lack of social support, unmet needs of accessible adolescent-friendly sexual health services, counseling, and age-appropriate information may contribute to poorer mental health. Therefore, addressing sexual and mental health concurrently could play an important role in addressing the overall well-being of young people. Future studies in diverse contexts are needed in order to achieve a deeper understanding of the concept of sexual health as understood by adolescents and its relation to psychological well-being. Such an understanding will also allow health care professionals work closely with adolescents to develop and test effective youth-friendly sexual and reproductive health interventions.

**Author Contributions:** Conceptualization, N.S.P., E.P. and K.H.; methodology, N.S.P., E.P. and K.H.; software, N.S.P., E.P.; validation, N.S.P., E.P and K.H.; formal analysis, N.S.P., E.P. and K.H.; writing—original draft preparation, N.S.P.; writing—review and editing, N.S.P., E.P., K.H., S.H., Z.M. and M.J.; visualization, N.S.P., E.P., K.H., S.H., Z.M. and M.J. supervision, E.P. and K.H. All authors have read and agreed to the published version of the manuscript.

**Funding:** This research received no external funding.

**Institutional Review Board Statement:** Not Applicable.

**Informed Consent Statement:** Not Applicable.

**Data Availability Statement:** The data extraction forms used and analyzed in the current study are available, upon reasonable request, from the corresponding author.

**Conflicts of Interest:** The authors declare no conflict of interest.

## Appendix A

**Table A1.** Scoping Review: Summary of the 12 Studies Reviewed.

| S # | Title, Author, Country, and Year | Population Sample and Age | Purpose of the Study | Study Design and Method | Main Findings |
|---|---|---|---|---|---|
| 1 | A Different Approach in Developing a Sexual Self-Concept Scale for Adolescents in Accra, Ghana<br>Authors: Biney, A. A. E.<br>Country: Ghana<br>Year: 2016 | Quantitative:<br>● A total of 196 adolescents (102 girls and 94 boys), 12–19 years, participated in the survey.<br>● 52% male, 48% females<br>Qualitative:<br>● 50 adolescents;<br>● 12–14-year olds;<br>● 15–19-year olds;<br>● 54% males and 46% female | To explore if there are significant relationships between adolescents' sexual self-concept and their sexual and mental health | Mixed Method: Quantitative:<br>● Survey: finalizing the SSC scale items and developing and validating the scale.<br>Qualitative:<br>● Focus group discussions and content analysis | Quantitative findings:<br>● The majority reported good mental health scores (mean = 25.5; halfway mark: 18).<br>Qualitative findings:<br>● Sexual fearlessness perceived as damaging to emotional well-being. Emotions of fear (or the lack of fear) associated with feelings of happiness, nervousness, and hopelessness. |
| 2 | Mental Well-being and Self-reported Symptoms of Reproductive Tract Infections among Girls: Findings from a Cross-sectional Study in an Indian Slum<br>Authors: Khopkar, S. A., Kulathinal, S., Virtanen, S. M., & Säävälä, M.<br>Country: India | 10–18-year-old adolescent girls (*n* = 85) | To assess the associations between socio-demographic variables, physical health indicators, and adolescent post-menarcheal girls' mental well-being. | Quantitative study: Cross-sectional personal interview survey | The mean and standard deviation of the mental well-being score (scale 0 to 12) were 8 and 3. Each post-menarcheal girl in the inner-city slum was classified as having a low (score 0 to 8) or high (score 9 to 12) score. A total of 36 girls had low scores, while 49 had high scores. The level of maturation gave an indication of potentially being related to worsening mental well-being scores. |
| | Year: 2017 | | | | Nearly every other post-menarcheal girl reported having experienced symptoms suggestive of reproductive tract infections during the last twelve months. |

**Table A1.** *Cont.*

| S # | Title, Author, Country, and Year | Population Sample and Age | Purpose of the Study | Study Design and Method | Main Findings |
|---|---|---|---|---|---|
| 3 | Emotional and Psychosocial Aspects of Menstrual Poverty in Resource-Poor Settings: A Qualitative Study of the Experiences of Adolescent Girls in an Informal Settlement in Nairobi Authors: Crichton, J., Okal, J., Kabiru, C. W., & Zulu, E. M. Country: Nairobi, Kenya Year: 2013 | Adolescent girls aged 12 to 17 years to ensure our sample reflected variations in age (12–14, 15–17 age groups) | To examine the impact of menstrual poverty on the emotional well-being of adolescent girls in an informal settlement in Nairobi, Kenya | Qualitative study purposive quota sampling open-ended interview questions 15 in-depth interviews (IDIs) and 10 focus group discussions (FGDs) A total of 87 girls participated in FGDs | Girls experienced psychosocial deprivations, including limited access to information and lack of emotional and practical support with menstruation from parents and family members. Lack of reliable access to menstrual products was a major cause of physical discomfort, embarrassment, anxiety, fear of being stigmatized and low mood. Participants used language, such as "feeling bad," feeling "stressed", or "fearful" and "wanting to cry", to describe the emotional distress. Negative feelings were associated with menstrual poverty and caused anxiety during school days. Hormone-related symptoms of fatigue and mood symptoms, including tension and depressed mood, are highly prevalent among menstruating girls regardless of social context or menstrual poverty. |
| 4 | Unmet medical care and sexual health counseling needs: a cross-sectional study among university students in Uganda Authors: Kyagaba, E., Asamoah, B. O., Emmelin, M., & Agardh, A. Country: Uganda Year: 2014 | $n = 1954$ students below the age of 24 56% male and 44% female | To investigate unmet medical care and sexual health counseling needs among the study population chosen (Ugandan university students) in order to see how these needs are associated with mental health, social capital, religion, and sexual behavior. | Quantitative study: self-administered questionnaire containing 132 items | The majority of students (81%) reported having good self-rated health, but 51% said they had unmet medical needs, and 26% reported unmet sexual health counseling needs. Students with high mental health scores (i.e., poor mental health, $p$-value < 0.001) who practiced inconsistent condom use ($p$-value 0.0059, $p$-value 0.006), who had experienced sexual coercion ($p$-value < 0.001), and who had poor self-rated health ($p$-value < 0.001) had a higher prevalence of both unmet medical care and sexual health counseling needs. The association between risky sexual behaviors among men and unmet sexual and reproductive health service needs explained by the fear of being stigmatized or punished for sexual activity when seeking care. Poor mental health is highly stigmatized and individuals who are perceived as having a low mental health status seem to be less willing to seek health care. |

**Table A1.** *Cont.*

| S # | Title, Author, Country, and Year | Population Sample and Age | Purpose of the Study | Study Design and Method | Main Findings |
|---|---|---|---|---|---|
| 5 | Adolescents' Responses to an Unintended Pregnancy in Ghana: A Qualitative Study Authors: Aziato, L., Hindin, M. J., Maya, E. T., Manu, A., Amuasi, S. A., Lawerh, R. M., & Ankomah, A. Country: Ghana Year: 2016 | 92 adolescents girls, aged 13–19 years | To investigate the experiences and perceptions of adolescents who have experienced a recent pregnancy and undergone a termination of pregnancy. To clarify if the sample had indeed experienced pregnancy | Qualitative study: A vignette-based focus group approach with fifteen FGDs | Adolescents reported that the characters in the vignettes would feel sadness, depression, and regret from unintended pregnancies. Most participants believed the parents of a pregnant adolescent in the vignette would not be happy about the pregnancy and the parents' potential reactions would range from sadness and annoyance to anger and abuse. Health care professionals are a source of stress as they are likely to be judgmental and disrespectful. |
| 6 | Menarche stories: reminiscences of college students from Lithuania, Malaysia, Sudan, and the United States. Authors: Joan, C. C., & Zittel, P. C. B Country: 26 Lithuanians, 27 Americans, 20 Malaysians, and 23 Sudanese Year: 1998 | 26 Lithuanian, 27 American, 20 Malaysian, and 23 Sudanese girls The Malaysian students were 19 to 20 years old The Sudanese women's average age was 20 years old | This study aims to understand and analyze the experience of first menstruation, emotional reaction, preparedness, sources of information about menstruation, changes in body image, and celebrations of this rite of passage. | Qualitative study: Female psychology students were invited to write the story of their first menstruation. | The most common emotions mentioned by the Malaysians were fear and embarrassment, followed closely by worry. The most common emotion mentioned by the Sudanese was fear; also common were anxiety, embarrassment, and anger. |
| 7 | Physical, Social, and Political Inequities Constraining Girls' Menstrual Management at Schools in Informal Settlements of Nairobi, Kenya. Authors: Girod, C., Ellis, A., Andes, K. L., Freeman, M. C., & Caruso, B. A. Country: Kenya Year: 2017 | Schoolgirls 6–11 post-menarchal girls in grades 6–8 | This study documents differences between girls' experience of menstruation at public schools (where the Kenyan government provides menstrual pads) and private schools (where pads are not provided) in two informal settlements of Nairobi, Kenya. | Qualitative study: focus group discussion (FGD) with girls | Girls experienced fear and anxiety due to harassment from male peers and had incomplete information about menstruation from teachers. Girls in every school had fear and anxiety about getting infections. They worried about negative health outcomes due to poor menstrual management, and they believed that urine splattering onto the vulva could cause urinary tract infections, gonorrhea, or infertility. |
| 8 | Adolescent perception of reproductive health care services in Sri Lanka. Authors: Agampodi, S. B., Agampodi, T. C., & Piyaseeli, U. K. D. Country: Sri Lanka Year: 2008 | 32 adolescents between 13 males and 19 females 17–19 years of age | The purpose of this study was to explore the perceived reproductive health problems, health-seeking behaviors, knowledge about available services and barriers to reach services among a group of adolescents in Sri Lanka in order to improve reproductive health service delivery. | Qualitative study: four focus group discussions | Psychological distresses due to various reasons and problems regarding the menstrual cycle and masturbation are the most common health problems. |

**Table A1.** *Cont.*

| S # | Title, Author, Country, and Year | Population Sample and Age | Purpose of the Study | Study Design and Method | Main Findings |
|---|---|---|---|---|---|
| 9 | The importance of a positive approach to sexuality in sexual health programs for unmarried adolescents in Bangladesh. Authors: van Reeuwijk, M., & Nahar, P. Country: Bangladesh. Year: 2013 | Young, unmarried adolescents of 12–18 years | To explore the mismatch that exists between what unmarried adolescents in Bangladesh experience, want and need with regard to their sexuality and what they receive from their society, which negatively impacts on their understanding of sexuality and their well-being. | Qualitative study: in-depth interviews, focus group discussion, observations, and content analysis | Many girls expressed worries and various misconceptions about the issue of virginity and were insecure about their ability to prove their own virginity. Boys were curious about masturbation and wet dreams and about the size and shape of the penis and duration of intercourse. Boys felt bad for having wet dreams and a number felt guilty after masturbating. |
| 10 | Adolescent and Parental Reactions to Puberty in Nigeria and Kenya: A Cross-Cultural and Intergenerational Comparison. Authors: Bello, B. M., Fatusi, O., Adepoju, O. E., Maina, W., Kabiru, C. W., Sommer, M., & Mmari, K. Country: Nigeria and Kenya Year: 2017 | Sixty-six boys and girls (aged 11 to 13 years) | To assess the reactions of adolescents and their parents to puberty in urban poor settings in two African countries Nigeria (Ile-Ife) in West Africa, and Kenya (Nairobi) in East Africa and compared the experiences of current adolescents to that of their parents' generation. | Qualitative study | Adolescents' reactions to puberty-related bodily changes varied from anxiety, shame, to pride, and an increased desire for privacy. |
| 11 | Factors impacting on menstrual hygiene and their implications for health promotion. Authors: Lahme, A. M., Stern, R., & Cooper Country: Zambia Year: 2018 | 51 respondents, aged 13–20 years | This paper explores the factors influencing the understanding, experiences and practices of menstrual hygiene among adolescent girls in Mongu District, Western Province of Zambia. | Explorative Qualitative study Six focus group discussions | Girls suffer from poor menstrual hygiene, originating from lack of knowledge, culture and tradition, and socio-economic and environmental constraints, leading to inconveniences, humiliation and stress. This leads to reduced school attendance and poor academic performance, or even dropouts, and ultimately infringes upon the girls' human rights. |
| 12 | Menstrual knowledge and practices of female adolescents in Vhembe district, Limpopo Province, South Africa Authors: Ramathuba, D. U. Country: South Africa Year: 2015 | 14–19 years 273 secondary school girls doing Grades 10–12 | This study sought to assess the knowledge and practices of secondary school girls towards menstruation in the Thulamela municipality of Limpopo Province, South Africa. | A quantitative descriptive study design | 73% of girls reported having fear and anxiety at the first experience of bleeding |

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
