# Peer review of "Associations between Developing Sexuality and Mental Health in Heterosexual Adolescents: Evidence from Lower- and Middle-Income Countries—A Scoping Review"

_adolescents, doi:10.3390/adolescents2020015_

Round 1
Reviewer 1 Report
The paper presents an interesting and pertinent objective that serves different purposes, including the possibility of identifying implications for practice and informing public policy.
The methodology adopted is robust and well explained, considering the research for an extended period and a wide set of databases.
The methodological process is clearly explained and described and the results consider the selected studies, with interesting identification and analysis of themes and sub-themes.
The fact that the research and analysis focus on a specific population group and on countries in particular contexts (LIMC) allows for more focused, detailed analysis and the identification of challenges that arise in these contexts, with a greater possibility of obtaining relevant results for intervention.
I also consider that some aspects could be improved or better justified:
- At the end of section 1.1. appears a set of sentences that do not belong to the text (it seems to be a review text of the article).
- It is not clear the criteria used to delimit, in age terms, the population under study (in fact, it seems to have selected adolescents and young adults, from 11 to 24 years old, which does not correspond to what is said in some parts of the article, including the title).
- Despite the reference in the limitations, the application of the exclusion criteria to the lesbian, bisexual, queer and transgender population is neither clear nor discussed.
Author Response
Reviewer |
Comments |
Correction |
Reviewer 1 |
At the end of section 1.1. appears a set of sentences that do not belong to the text (it seems to be a review text of the article). |
This has been removed form the article |
Reviewer 1 |
It is not clear the criteria used to delimit, in age terms, the population under study (in fact, it seems to have selected adolescents and young adults, from 11 to 24 years old, which does not correspond to what is said in some parts of the article, including the title). |
Thank you for the comment
We have now corrected the age as adolescent population (11-19 years) |
Reviewer 1 |
Despite the reference in the limitations, the application of the exclusion criteria to the lesbian, bisexual, queer and transgender population is neither clear nor discussed. |
Under section 2.3, we have mentioned that we have excluded the LGBTQ population, as the issues of sexual health and mental health could be different in this population |
Reviewer 2 Report
The work presented is of enormous interest and relevance. Although the development of sexuality has been extensively researched, its relationship to mental health has been little addressed in the literature.
Here are some aspects that should be taken into account in order to improve the work:
- The title should state that the article is a review.
- The search is too old. Review articles should be from the last 6 months at the latest. In this case it is a review carried out 4 years ago.
Author Response
Reviewer 2 |
The title should state that the article is a review. |
Thank you for this comment, we have now revised the title and added it’s a scoping review |
Reviewer 2 |
The search is too old. Review articles should be from the last 6 months at the latest. In this case it is a review carried out 4 years ago. |
We have re-run the search from 2019-2021. But we are unable to find any new article that qualify for our eligibility criteria. This is revised in the paper |
Reviewer 3 Report
In the abstract: Results section, you have listed number 4 twice.
Introduction:
It may be a paraphrase, but the WHO definition of sexuality feels close to a direct quote. Either paraphrase or directly quote, as it informs your study.
It would further be beneficial to include why you have highlighted gender in the definition - this shows up again in the discussion but not elsewhere explicitly in the paper.
The end of the introduction before the Aim, there is a 6 sentences that are copied from author instructions.
Methods:
Study Eligibility Criteria, you have noted populations you did not include but not why.
Results:
3rd paragraph of first section has a sentence, "However, it was not possible to perform a gender analysis as gender- specific results were not reported in the primary studies." What does "gender- specific results" mean in this context? In the discussion more studies with girls than boys is discussed, is this what you mean? Or are you talking about the difference between transgender and cisgender studies?
I especially appreciated the direct quotes and notes of themes in the qualitative papers, I did not see any specific statistics or data from the quantitative studies referenced in results. It would be helpful to know what if there were any noteworthy results from those studies as well.
Great job, this was a very interesting paper!
Author Response
Reviewer 3 |
In the abstract: Results section, you have listed number 4 twice. |
This has been removed |
Reviewer 3 |
It may be a paraphrase, but the WHO definition of sexuality feels close to a direct quote. Either paraphrase or directly quote, as it informs your study. |
We have now added sexuality definition as a quote |
Reviewer 3 |
It would further be beneficial to include why you have highlighted gender in the definition - this shows up again in the discussion but not elsewhere explicitly in the paper. |
Under section 1.1, we have now added that “We have focused on gender as in many Low- and Middle-Income Countries (LMICs) traditional gender roles shape the way adolescent girls and boys explore their sexualities. “ |
Reviewer 3 |
The end of the introduction before the Aim, there is 6 sentences that are copied from author instructions. |
This has been removed from the article |
Reviewer 3 |
Study Eligibility Criteria, you have noted populations you did not include but not why. |
We have now added the reason for exclusion under section 2.3 |
Reviewer 3 |
3rd paragraph of first section has a sentence, "However, it was not possible to perform a gender analysis as gender- specific results were not reported in the primary studies." What does "gender- specific results" mean in this context? In the discussion more studies with girls than boys is discussed, is this what you mean? Or are you talking about the difference between transgender and cisgender studies? |
Keeping this suggestion in mind, we have now explicitly mentioned that gender specific means girls and boys |
Reviewer 3 |
I especially appreciated the direct quotes and notes of themes in the qualitative papers, I did not see any specific statistics or data from the quantitative studies referenced in results. It would be helpful to know what if there were any noteworthy results from those studies as well. |
Thanks for this suggestion Unfortunately, we mostly had qualitative papers and not much statistics to report in results section. |
Round 2
Reviewer 2 Report
The modifications made by the authors meet the demands made by this reviewer.